# Opposing effects of T cell receptor signal strength on CD4 T cells responding to acute versus chronic viral infection

**Marco Künzli[1], Peter Reuther[2], Daniel D Pinschewer[2], Carolyn G King[1]***

[1]Immune Cell Biology Laboratory, Department of Biomedicine, University of Basel, University Hospital Basel, Basel, Switzerland; [2]Division of Experimental Virology, Department of Biomedicine – Haus Petersplatz, University of Basel, Basel, Switzerland

**Abstract** A hallmark of adaptive immunity is CD4 T cells' ability to differentiate into specialized effectors. A long-standing question is whether T cell receptor (TCR) signal strength can dominantly instruct the development of Th1 and T follicular helper (Tfh) cells across distinct infectious contexts. We characterized the differentiation of murine CD4 TCR transgenic T cells responding to altered peptide ligand lymphocytic choriomeningitis viruses (LCMV) derived from acute and chronic parental strains. We found that TCR signal strength exerts opposite and hierarchical effects on the balance of Th1 and Tfh cells responding to acute versus persistent infection. TCR signal strength correlates positively with Th1 generation during acute but negatively during chronic infection. Weakly activated T cells express lower levels of markers associated with chronic T cell stimulation and may resist functional inactivation. We anticipate that the panel of recombinant viruses described herein will be valuable for investigating a wide range of CD4 T cell responses.

## Introduction

Following infection or vaccination, antigen-specific T cells undergo clonal expansion and differentiation into effector cells with specialized functions. This process begins with T cell receptor (TCR) recognition of peptide/major histocompatibility complex (pMHC) on antigen presenting cells (APCs) and is further modulated by cytokines and costimulatory molecules (*Zhu et al., 2010*; *Davis et al., 1998*). Viral infection induces the early bifurcation of CD4 T cells into Th1 and T follicular helper (Tfh) cells. Th1 cells potentiate CD8 T cell and macrophage cytotoxicity, whereas Tfh cells support antibody production by providing survival and proliferation signals to B cells (*Zhu et al., 2010*; *Crotty, 2011*).

Although the cumulative strength of interaction between TCR and pMHC has a clear impact on T cell expansion and fitness, its influence on the acquisition of Th1 and Tfh cell fates is controversial (*Keck et al., 2014*; *Kotov et al., 2019*; *Snook et al., 2018*; *DiToro et al., 2018*; *Krishnamoorthy et al., 2017*; *Tubo et al., 2013*; *Ploquin et al., 2011*; *Fazilleau et al., 2009*; *Vanguri et al., 2013*). An essential role for TCR signal was implicated in a study assessing the phenotype of progeny derived from individual, TCR transgenic (tg) T cells responding during infection (*Tubo et al., 2013*). The authors observed that distinct TCRs induced reproducible and biased patterns of Th1 and Tfh phenotypes. Although earlier reports suggested that Tfh cell differentiation requires high TCR signal strength, recent work supports the idea that Tfh cells develop across a wide range of signal strengths, while increasing TCR signal intensity favors Th1 generation (*Keck et al., 2014*; *Kotov et al., 2019*; *Snook et al., 2018*; *DiToro et al., 2018*; *Krishnamoorthy et al., 2017*; *Tubo et al., 2013*; *Ploquin et al., 2011*; *Fazilleau et al., 2009*). A central difficulty in reconciling these findings is the use of different TCR tg systems as well as

*For correspondence:
Carolyn.King@unibas.ch

immunization and infection models that may induce distinct levels of costimulatory and inflammatory signals known to influence T cell differentiation. Although existing reports suggest that persistent TCR signaling drives a shift toward Tfh differentiation during chronic infection, whether this outcome can be modulated by TCR signal strength has not been examined (*Oxenius et al., 1998*; *Fahey et al., 2011*; *Crawford et al., 2014*).

The impact of TCR signal strength on CD4 T cell differentiation in vivo has been historically challenging to address. The use of MHC-II tetramers to track endogenous polyclonal T cell responses does not adequately detect low affinity T cells that can comprise up to 50% of an effector response in autoimmune or viral infection settings (*Sabatino et al., 2011*). TCR tg models paired with a panel of ligands with varying TCR potency have been informative, but only a handful of MHC-II restricted systems exist (*Corse et al., 2010*; *Huseby et al., 2006*). To bypass these limitations, the generation of novel tg/retrogenic TCR strains or recombinant pathogen strains is required (*Keck et al., 2014*; *Vanguri et al., 2013*; *Gallegos et al., 2016*; *Kim et al., 2013*). To our knowledge, this approach has not yet been used to modify a naturally occurring CD4 T cell epitope of an infectious agent. To test the impact of TCR signal strength across different types of infectious contexts, we generated a series of lymphocytic choriomeningitis virus (LCMV) variants by introducing single amino acid mutations into the GP61 envelope glycoprotein sequence and expressing them from both acute and chronic parent strains. These strains were used to assess the dynamics and differentiation of SMARTA T cells, a widely used TCR tg mouse line that mirrors the endogenous, immunodominant CD4 T cell response to LCMV (*Oxenius et al., 1998*). We observed that depending on the infection setting, TCR signal strength has opposing effects on the balance between Th1 and Tfh cell differentiation. In an acute infection, strong TCR signals preferentially induce Th1 effectors, whereas weak TCR signals shift the balance toward Tfh effectors. In contrast, strong T cell activation during chronic infection induces Tfh cell differentiation while more weakly activated T cells are biased to differentiate into Th1 cells.

## Results

### Generation and viral fitness of GP61 LCMV variants

To generate recombinant LCMV variants, we first screened a panel of altered peptide ligands (APLs) with single amino acid mutations in the LCMV derived GP61 peptide. Using the early activation marker CD69 as a proxy for TCR signal strength, we identified 75 APLs for the SMARTA TCR tg line (*Figure 1—figure supplement 1*, *Figure 1A,B*, *Supplementary file 1*). For several of these APLs, we additionally assessed CD69, CD25, and IRF4 expression in the presence of interferon (IFN) type-I blocking antibody to ensure that T cell activation was driven by TCR signaling and not secondarily through the IFN-receptor (*Figure 1—figure supplement 2*; *Shiow et al., 2006*). Blocking interferon-a receptor (IFNAR) had no effect on CD69 expression at 6 hr after activation, while a CD25 and IRF4 were modestly decreased in SMARTA T cells activated with low-dose APL, suggesting that the activation hierarchy of these APLs is mainly a result of differences in TCR signal strength. Next, using an MHCII-binding prediction tool (http://tools.iedb.org/mhcii/), we selected 12 of these APLs with favorable MHCII-binding predictions that covered a wide range of T cell activation potential to generate recombinant variant viruses using site-directed PCR mutagenesis (*Flatz et al., 2006*). Five APL-encoding sequences were successfully introduced into the genomes of both LCMV Armstrong (Armstrong variants) and Clone-13 (Clone-13 variants), the latter of which contains a mutation in the polymerase gene L that enhances the replicative capacity of the virus, enabling viral persistence (*Bergthaler et al., 2010*; *Sullivan et al., 2011*). To exclude a potential impact of differential glycoprotein-mediated viral tropism on CD4 T cell differentiation, we equipped both the Armstrong- and Clone-13-based viruses with the identical glycoprotein of the WE strain and introduced the epitope mutations therein (resulting viruses referred to as rLCMV Armstrong and rLCMV Clone-13, respectively) (*Bergthaler et al., 2007*). Of these viruses, two variants, V71S and Y72F (EC$_{50}$ ~0.1 and 1 μM, respectively) demonstrated comparable viral fitness in vitro and in vivo, with all three Clone-13 variants persisting in the blood, spleen, and kidney throughout the 21-day observation period (*Figure 1C–E*). To additionally rule out any contribution of endogenous CD4 T cells to viral control, we also determined the viral load in DBA/2 mice which are unable to present the GP61 epitope. Here, again we observed comparable viral load across all organs at 21 days post infection and

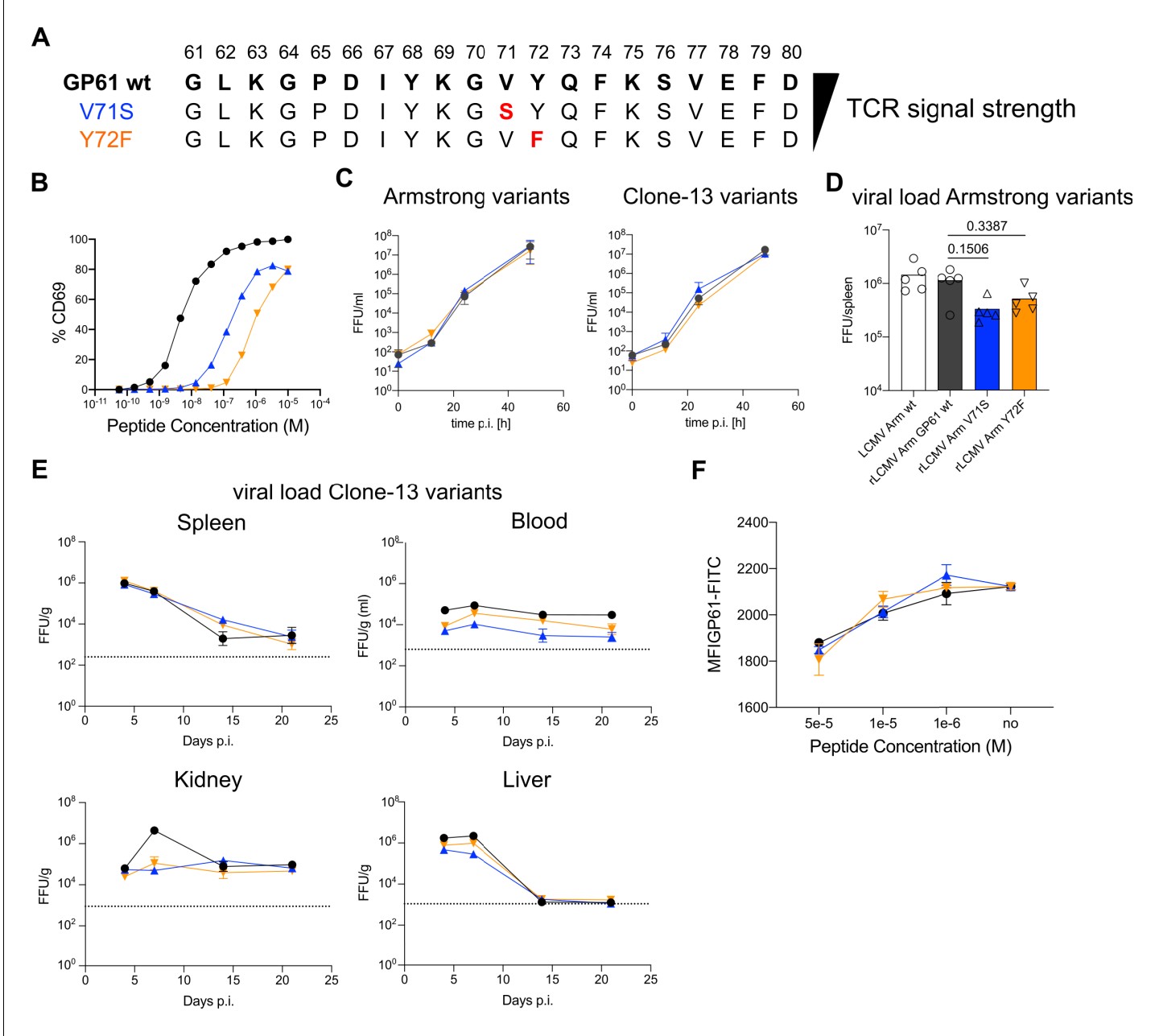

**Figure 1.** Generation and viral fitness of GP61 lymphocytic choriomeningitis virus (LCMV) variants. (**A**) Scheme of GP61 wt and altered peptide ligand (APL) sequences with mutations highlighted in red ordered hierarchically according to T cell receptor (TCR) signal strength. (**B**) Peptide dose–activation curves of overnight cultured SMARTA cells with peptide pulsed splenocytes using the percentage of $CD69^+$ SMARTA cells as a readout for activation. $EC_{50}$ values are ~5 nM for GP61 wt, ~0.1 μM for V71S, and ~1 μM for Y72F. (**C**) In vitro growth kinetics depicting the viral load in the culture medium (focus forming units [FFU]/ml) of GP61 wt or V71S and Y72F variants of Armstrong (left) and Clone-13 (right) variant infection on BHK21 cells over time. Data are displayed as mean ± SD. (**D**) Early splenic viral load day 3 post infection (p.i.) in Armstrong variants. Bars represent the mean and symbols represent individual mice. (**E**) Viral load (FFU) in indicated organs per gram tissue over time in C57BL/6 mice. The dotted line represents the limit of detection. Data are displayed as mean ± SEM of 7–10 samples. (**F**) Peptide dose–response curves depicting the out-competition of the GP61 FITC signal by unlabeled GP61 wt or APLs on $B220^+$ B cells. Data are displayed as mean ± SD of two to three technical replicates. Data represent one of n = 2 independent experiments (B, D, F) or pooled data from n = 2 independent experiments (C, E).

The online version of this article includes the following source data and figure supplement(s) for figure 1:

**Source data 1.** Generation and viral fitness of GP61 lymphocytic choriomeningitis virus (LCMV) variants.

**Figure supplement 1.** GP61 altered peptide ligand (APL) screening.

**Figure supplement 2.** Interferon (IFN) type-I signaling exerts minor effects on the expression of activation markers in vitro.

*Figure 1 continued on next page*

*Figure 1 continued*

**Figure supplement 3.** Viral load in organs and blood of DBA/2 mice.

in the blood over time (*Figure 1—figure supplement 3*). Finally, as GP$_{67-77}$, the minimal binding epitope of GP61, can bind to MHC in different registers, we considered the possibility that V71 and Y72 might act as MHC anchor residues as opposed to TCR contact residues (*Homann et al., 2007*). To control for this, we performed an out-competition assay with invariant chain knockout splenocytes (*Liu et al., 2002*). Here, we observed comparable binding of APL peptides to MHC-II, indicating that reduced activation of SMARTA T cells is unlikely to be a result of V71 and Y72 acting as crucial MHC anchor residues (*Figure 1F*). Taken together, these data demonstrate the development of a novel tool to examine the impact of TCR signal strength on SMARTA T cells activated by either acute or chronic viral infection.

To assess the impact of TCR signal strength during acute viral infection, SMARTA T cells were transferred into congenic recipients followed by infection with rLCMV Armstrong GP61 wt, V71S, or Y72F. All LCMV variants were capable of inducing SMARTA T cell expansion at day 10 post infection (p.i.), and a direct correlation between TCR signal strength and the number of SMARTA T cells recovered was observed (*Figure 2A*). In contrast, expansion of endogenous LCMV nucleoprotein (NP)-specific as well as antigen-experienced CD44$^+$ T cells was similar across all three viral strains (*Figure 2A*). The expansion hierarchy among the viruses was maintained >30 days after LCMV infection (*Figure 2A*).

We next examined the phenotype of SMARTA T cells, focusing our analyses on effector cells due to the impaired generation of Tfh memory by SMARTA T cells (*Künzli et al., 2020*). As the Y72F variant induced very few effector cells, we excluded this strain from further investigation. Consistent with previous reports, strong T cell stimulation induced a larger proportion of Ly6c$^+$ Th1 effectors, whereas the proportion of Tfh effectors was decreased (*Figure 2B–D*; *Keck et al., 2014*; *Krishnamoorthy et al., 2017*; *Tubo et al., 2013*; *Ploquin et al., 2011*). Differences in SMARTA T cell expansion and the proportion of CD25$^+$ Th1 precursor cells were observed as early as day 4 after infection, preceding germinal center responses in both viruses (*Figure 2—figure supplements 1–3*).

In contrast, the ratio of Th1 and Tfh effector cells generated by host NP-specific T cells was consistent across all viral strains, providing an internal control for the comparable ability of these viruses to induce endogenous T cell responses (*Figure 2D*, *Figure 2—figure supplement 4*). Expression of folate receptor 4, an alternative marker for Tfh cell identification, was additionally used in combination with PD1 to discriminate the Tfh cell compartment and demonstrated a decreased proportion of Tfh cells activated by strong compared to weak TCR stimulation (*Figure 2—figure supplement 5*; *Künzli et al., 2020*; *Iyer et al., 2013*). Accordingly, the expression of Bcl6 and T-bet, lineage defining transcription factors for Tfh and Th1 cells, respectively, revealed a mild but significant trend toward increased Tbet and decreased Bcl6 in response to strong stimulation (*Figure 2E*). Importantly, Bcl6 expression was higher on Tfh compared to Th1 effectors, with no differences observed between strong and weak stimulation (*Figure 2F*, *Figure 2—figure supplement 6*). These data indicate that TCR signal strength is unlikely to exert a qualitative impact on these subsets. Notably, we did not observe any impact of TCR signal strength on the development of PSGL1$^{hi}$Ly6c$^{lo}$ T cells, previously reported to be a less differentiated population of Th1 effectors (*Figure 2—figure supplement 7*; *Künzli et al., 2020*; *Marshall et al., 2011*). In sum, TCR signal strength positively correlates with an increased ratio of Th1 to Tfh effectors during acute LCMV infection.

In contrast to acute LCMV infection, SMARTA T cells responding to chronic LCMV preferentially adopt a Tfh effector phenotype (*Fahey et al., 2011*; *Crawford et al., 2014*). The impact of TCR signal strength within this context has not been determined, although affinity diversity among endogenous T cells is reportedly similar between acute and chronic LCMV infection (*Andargachew et al., 2018*). To directly assess the impact of TCR signal strength during chronic infection, we transferred SMARTA T cells into congenic recipients followed by infection with rLCMV Clone-13 expressing either GP61 wt, V71S, or Y72F. As an additional control, we infected mice with rLCMV Armstrong which induced a similar expansion of SMARTA, NP-specific, and CD4$^+$CD44$^+$ T cells as its Clone-13 counterpart (*Figure 3—figure supplement 1*). Consistent with the results from acute infection,

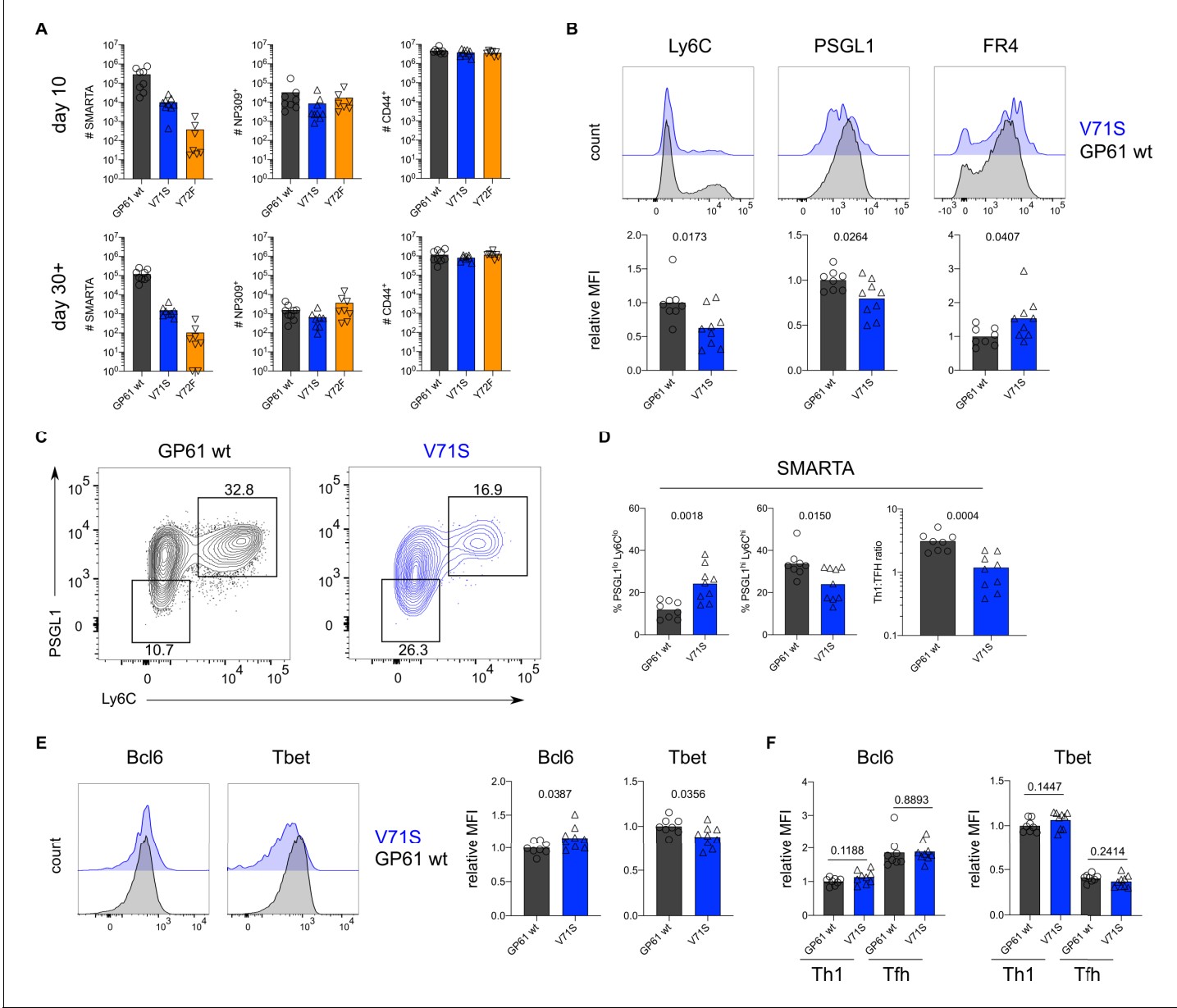

**Figure 2.** T cell receptor (TCR) signal strength positively correlates with Th1 cell differentiation during lymphocytic choriomeningitis virus (LCMV) Armstrong variant infection. (**A**) Number of SMARTA (left), NP309[+] (middle), and CD44[+] cells (right) 10 days (top) or >30 days (bottom) post infection (p. i.). (**B**) Histograms (top) and relative mean fluorescence intensity (MFI) (bottom) of indicated phenotypic markers in the SMARTA compartment 10 days p.i. (**C**) Identification of Th1 (Ly6C[hi]PSGL1[hi]) and T follicular helper (Tfh) (Ly6C[lo]PSGL1[lo]) subset in the SMARTA compartment by flow cytometry 10 days p.i. (**D**) Proportion of Tfh (left), Th1 cells (middle), and the Th1:Tfh ratio (right) of the SMARTA compartment 10 days p.i. (**E**) Histograms (left) and relative MFI (right) of Bcl6 and Tbet expression in the SMARTA compartment 10 days p.i. (**F**) Bcl6 and Tbet MFI in SMARTA Th1 and Tfh subsets. Data are pooled from n = 2 independent experiments with seven to nine samples per group. Bars represent the mean and symbols represent individual mice. Significance was determined by unpaired two-tailed Student's t-tests.

The online version of this article includes the following source data and figure supplement(s) for figure 2:

**Source data 1.** T cell receptor (TCR) signal strength positively correlates with T follicular helper (Tfh) cell differentiation during lymphocytic choriomeningitis virus (LCMV) Armstrong variant infection.

**Figure supplement 1.** SMARTA cell numbers 4 days post infection (p.i.).

**Figure supplement 2.** T cell receptor (TCR) signal strength positively correlates with early Th1 cell differentiation during lymphocytic choriomeningitis virus (LCMV) Armstrong variant infection.

**Figure supplement 3.** Germinal center B cell differentiation 4 days post infection (p.i.) is unaffected by lymphocytic choriomeningitis virus (LCMV) Armstrong variant infection.

*Figure 2 continued on next page*

*Figure 2 continued*

**Figure supplement 4.** The endogenous nucleoprotein (NP)-specific CD4 response is unaffected by lymphocytic choriomeningitis virus (LCMV) Armstrong variant infection.

**Figure supplement 5.** Alternative gating strategy to identify SMARTA T follicular helper (Tfh) cells.

**Figure supplement 6.** T cell receptor (TCR) signal strength does not impact Bcl6 and Tbet expression in Ly6C$^{lo}$ Th1 SMARTA cells.

**Figure supplement 7.** T cell receptor (TCR) signal strength does not impact the generation of Ly6C$^{lo}$ Th1 SMARTA cells.

SMARTA T cell numbers at day 7 p.i. positively correlated with TCR signal strength, while the expansion of NP-specific and CD4$^+$CD44$^+$ T cells was similar in response to all three Clone-13 variants (*Figure 3A*). Importantly, infection with rLCMV Clone-13 Y72F induced approximately twofold more SMARTA T cell effectors compared to acute infection, allowing for a thorough investigation of T cells responding to this very weak potency variant (*Figure 3A*, *Figure 2A*).

With respect to T cell phenotype, strong TCR stimulation during rLCMV Clone-13 GP61 wt infection shifted the balance toward Tfh effector cell differentiation when compared to strong TCR stimulation in the context of acute infection (*Figure 3—figure supplement 2*). Unexpectedly, and in contrast to the Armstrong variants, weaker TCR signaling during Clone-13 variant infection resulted in increased proportions of both PSGL1$^{hi}$Ly6c$^{hi}$ and PSGL1$^{hi}$Ly6c$^{lo}$ Th1 cells with the weakest variant, Y72F, generating the highest proportion of Th1 effectors (*Figure 3B–D*, *Figure 3—figure supplement 3*). The shift toward Th1 effectors in response to lower TCR signal strength is unlikely to be due to differences in antigen load as all variants sustained high viral titers in the kidneys at day 7 p.i. (*Figure 3—figure supplement 4*). In addition, although the viral titer of intermediate potency variant V71S was slightly decreased compared to GP61 wt and Y72F infection, NP-specific CD4 T cells exhibited a similar ratio of Th1 to Tfh effectors across all three infections (*Figure 3—figure supplement 5*). Similar to LCMV Armstrong infection, stronger Th1 differentiation was also observed at early time points after infection with Clone-13 variant viruses although no differences in germinal center B cell kinetics were observed (*Figure 3—figure supplements 6–9*). In contrast to LCMV Armstrong infection, however, CD25 expression was only weakly and uniformly expressed in all Clone-13 variants at day 4 after infection (*Figure 3—figure supplement 10*).

To determine whether the differential impact of TCR signal strength in Clone-13 infection might be partially due to antigen dose, we next examined the impact of lowering the antigen load while maintaining the inflammatory environment, mixing Clone-13 GP61 wt or Clone-13 Y72F at a 1:10 ratio with an rLCMV Clone-13 virus lacking the GP66 epitope (ΔGP66). In response to lowered antigen load, strongly stimulated SMARTA T cells expanded less and generated a higher proportion of Th1 effectors at day 7 p.i. while the expansion and differentiation outcome of weakly activated T cells was antigen load independent (*Figure 3—figure supplement 10*). Taken together, these data reveal that TCR signal strength differentially modulates T cell fate acquisition according to the infectious context and that biased Tfh differentiation during Clone-13 GP61 wt infection may be at least partially due to high antigen load.

During Clone-13 infection, T cells start to upregulate inhibitory surface markers associated with chronic activation, a state often referred to as 'exhaustion' (*Crawford et al., 2014*; *Dong et al., 2019*; *Mou et al., 2013*; *Jean Bosco et al., 2018*). To understand if TCR signal strength impacts the expression of these markers, we analyzed SMARTA T cells responding to Clone-13 GP61 wt and variant viruses at day 14. T cells responding to strong TCR signals expressed the highest levels of both PD1 and Lag3, two well characterized co-inhibitory receptors (*Figure 4A–B*; *Crawford et al., 2014*; *Dong et al., 2019*). SMARTA T cells co-expressing both PD1 and Lag3 were most abundant following Clone-13 GP61 wt infection and decreased in response to Clone-13 variant infection (*Figure 4C–D*). Although the viral load was decreased in Clone-13 variant infections at this time point, the basal activation of CD4$^+$CD44$^+$ T cells was equivalent across all three strains and clearly above the recombinant LCMV Armstrong control (*Figure 4—figure supplements 1–2*). Next, we examined the expression of TOX, a transcription factor involved in the adaptation of CD8 T cells to chronic infection (*Yao et al., 2019*; *Alfei et al., 2019*; *Khan et al., 2019*; *Scott et al., 2019*; *Seo et al., 2019*). In response to acute infection, SMARTA Tfh cells expressed higher levels of TOX compared to Th1 cells, consistent with an earlier study highlighting the importance of TOX for Tfh cell development (*Figure 4—figure supplement 3*; *Xu et al., 2019*). In contrast, TOX expression during rLCMV Clone-13 wt infection was most highly upregulated by Th1 effectors (*Figure 4—figure supplement*

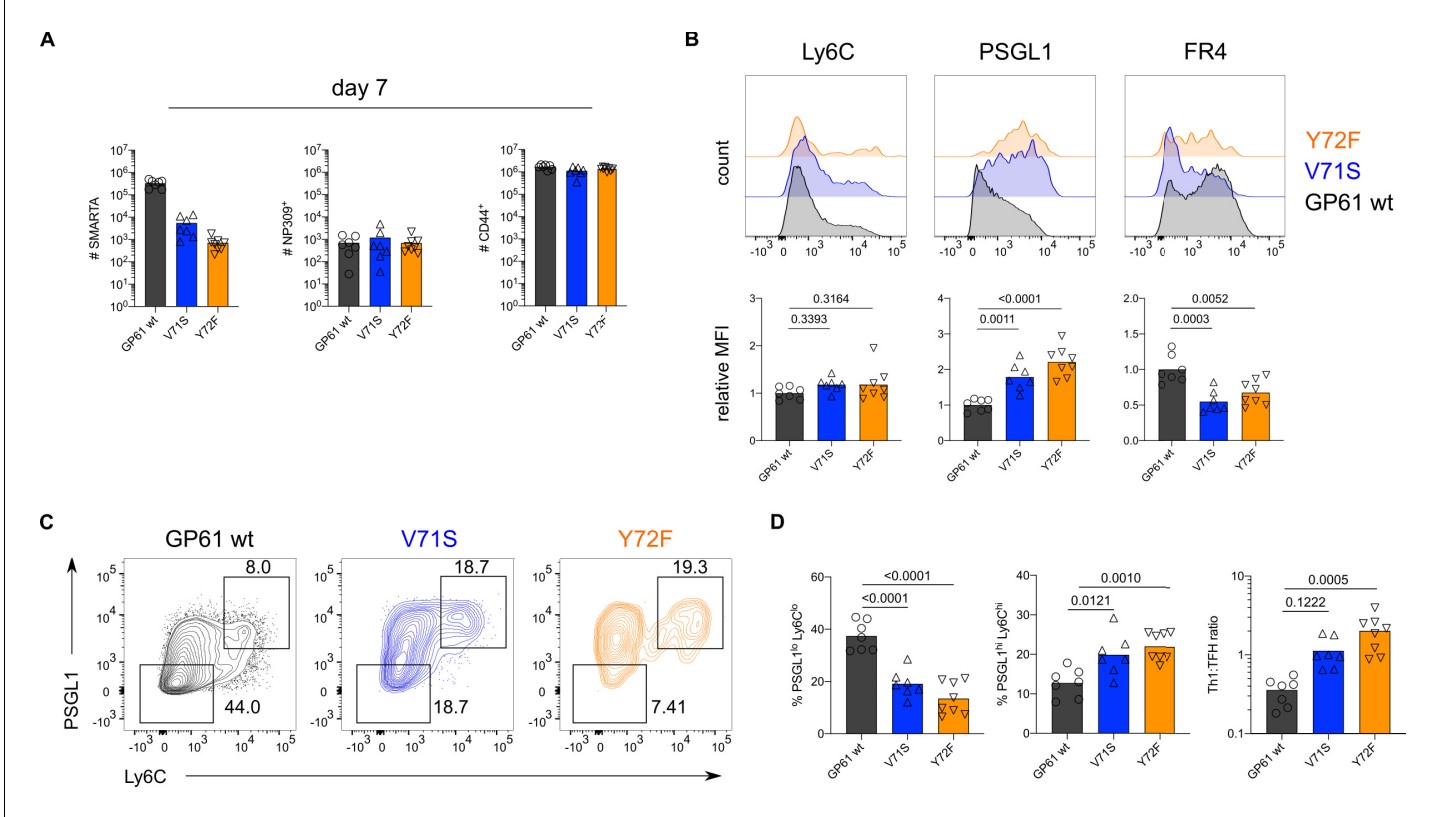

**Figure 3.** T cell receptor (TCR) signal strength positively correlates with T follicular helper (Tfh) cell differentiation during lymphocytic choriomeningitis virus (LCMV) Clone-13 variant infection. Spleens were harvested 7 days after infection with LCMV Clone-13 variants. (**A**) Number of SMARTA (left), NP309+ (middle), and CD44+ cells (right). (**B**) Histograms (top) and relative mean fluorescence intensity (MFI) (bottom) of indicated phenotypic markers in the SMARTA compartment. (**C**) Identification of Th1 (Ly6Chi PSGL1hi) and Tfh (Ly6Clo PSGL1lo) subset in the SMARTA compartment by flow cytometry. (**D**) Proportion of Tfh (left), Th1 cells (middle), and the Th1:Tfh ratio (right) of the SMARTA compartment. Data are pooled from n = 2 independent experiments with seven to eight samples per group. Bars represent the mean and symbols represent individual mice. Significance was determined by one-way ANOVA followed by Tukey's post-test.

The online version of this article includes the following source data and figure supplement(s) for figure 3:

**Source data 1.** IT cell receptor (TCR) signal strength positively correlates with T follicular helper (Tfh) cell differentiation during lymphocytic choriomeningitis virus (LCMV) Clone-13 variant infection..

**Figure supplement 1.** Lymphocytic choriomeningitis virus (LCMV) Armstrong and Clone-13 infection induce similar expansion of CD4 compartments.

**Figure supplement 2.** Lymphocytic choriomeningitis virus (LCMV) Clone-13 infection results in a T follicular helper (Tfh)-based differentiation of SMARTA cells.

**Figure supplement 3.** T cell receptor (TCR) signal strength negatively correlates with Ly6Clo Th1 differentiation during lymphocytic choriomeningitis virus (LCMV) Clone-13 variant infection.

**Figure supplement 4.** High viral load in kidneys 7 days post infection in all three viruses.

**Figure supplement 5.** The endogenous nucleoprotein (NP)-specific CD4 response is unaffected by lymphocytic choriomeningitis virus (LCMV) Clone-13 variant infection.

**Figure supplement 6.** SMARTA cell numbers 4 and 14 days post infection (p.i.).

**Figure supplement 7.** T cell receptor (TCR) signal strength negatively correlates with early Th1 cell differentiation during lymphocytic choriomeningitis virus (LCMV) Clone-13 variant infection.

**Figure supplement 8.** Th1 bias is maintained in Clone-13 variants 14 days post infection (p.i.).

**Figure supplement 9.** Germinal center B cell differentiation is unaffected by lymphocytic choriomeningitis virus (LCMV) Clone-13 variant infection.

**Figure supplement 10.** Antigen load exerts different effects on strongly versus weakly activated SMARTA T cells.

*3*). In line with the expression of PD1 and Lag3, TOX was decreased on SMARTA T cells responding to rLCMV Clone-13 variant viruses, despite being comparably induced on CD4+CD44+ T cells (*Figure 4E*, *Figure 4—figure supplement 4*). TOX was recently demonstrated to be important for the survival of stem-like TCF1+ CD8 T cells that accumulate during chronic LCMV (*Khan et al., 2019*;

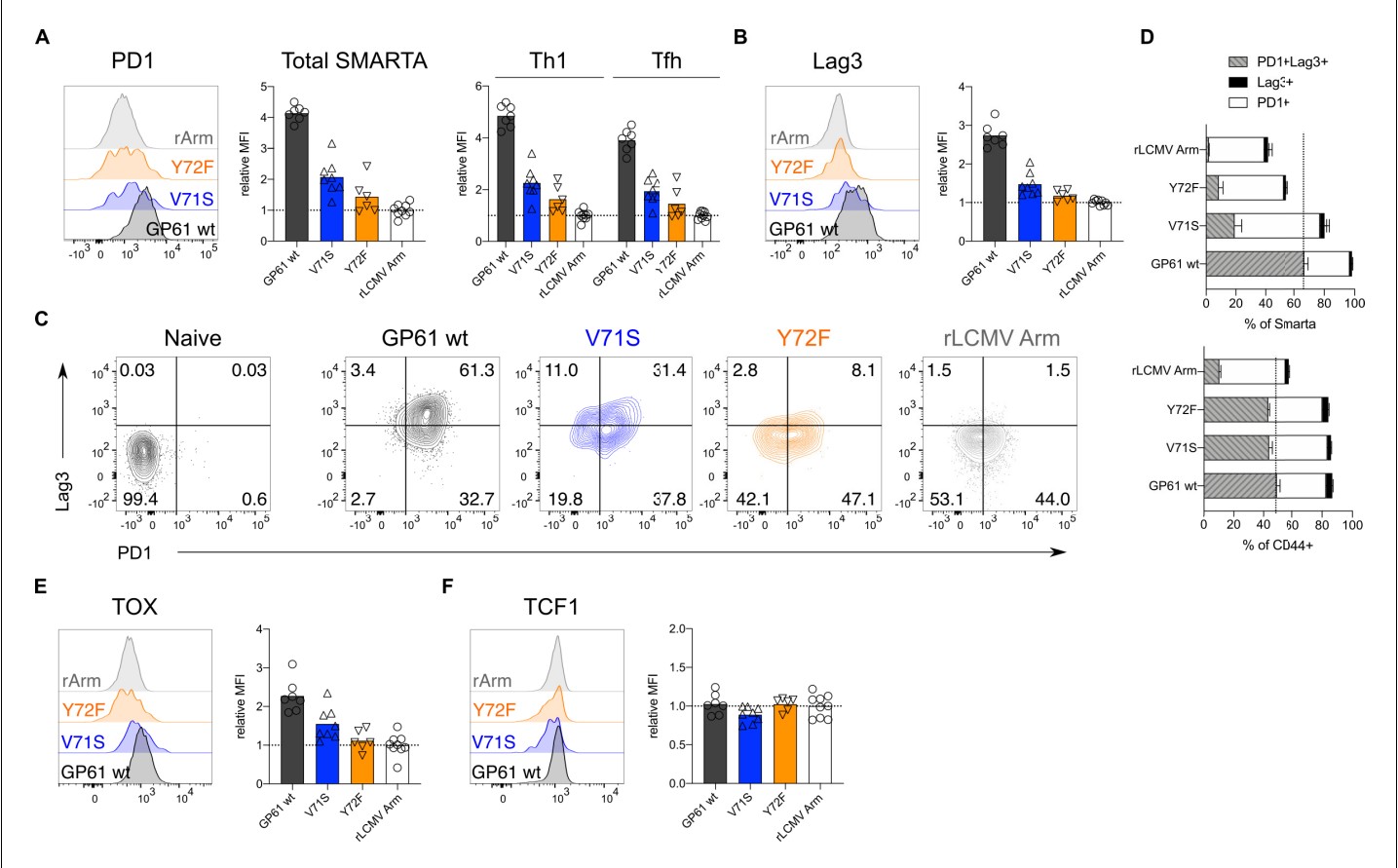

**Figure 4.** Increased T cell receptor (TCR) signal strength induces expression of markers associated with chronic T cell stimulation. Spleens were harvested 14 days after infection with lymphocytic choriomeningitis virus (LCMV) Clone-13-based variants. (**A**) Histograms (left) and relative mean fluorescence intensity (MFI) (right) of PD1 in the total SMARTA compartment (left) or SMARTA Th1 and T follicular helper (Tfh) subsets (right). (**B**) Histograms (left) and relative MFI (right) of Lag3 in the SMARTA compartment. (**C**) Identification of PD1+Lag3+ SMARTA cells by flow cytometry compared to naïve CD62L+ CD44− CD4 T cells from an uninfected mouse. (**D**) Quantification of PD1+Lag3+ SMARTA cells in the SMARTA (top) or CD44+ (bottom) compartment. (**E**) Histogram (left) and relative MFI (right) of TOX in the SMARTA compartment. (**F**) Histogram (left) and relative MFI (right) of TCF1 in the SMARTA compartment. Data are pooled from n = 2 independent experiments with six to nine samples per group. Bars represent the mean and symbols represent individual mice. Significance was determined by one-way ANOVA followed by Tukey's post-test.

The online version of this article includes the following source data and figure supplement(s) for figure 4:

**Source data 1.** Increased T cell receptor (TCR) signal strength induces expression of markers associated with chronic T cell stimulation.
**Figure supplement 1.** Viral load in kidneys 14 days post infection.
**Figure supplement 2.** Similar activation marker expression of CD4+ CD44+ T cells across all viruses.
**Figure supplement 3.** T cell receptor (TCR) signal strength impacts TOX expression in SMARTA Th1 and T follicular helper (Tfh) compartments.
**Figure supplement 4.** TOX expression is not affected in CD44+ CD4+ T cells by lymphocytic choriomeningitis virus (LCMV) Clone-13 variant infection.
**Figure supplement 5.** T cell receptor (TCR) signal strength does not impact TCF1 expression.

*Im et al., 2016*; *Utzschneider et al., 2016*). Given the transcriptional similarities of TCF1+ CD8 T cells and Tfh cells, we wondered if TCF1 would be similarly regulated by TCR signal strength following Clone-13 variant infection (*Vella et al., 2017*). Here we observed that unlike TOX expression, TCF1 is similarly expressed by T cells responding to all three rLCMV Clone-13 variants (*Figure 4F*, *Figure 4—figure supplement 5*), indicating that TCF1 expression is likely to be maintained independently of TCR signals.

The results of this study highlight the differential impact of TCR signal strength in shaping CD4 T cell fate according to the infection context. By systematically comparing the differentiation of TCR tg T cells responding to variant ligands in two distinct infection models, we demonstrate that the impact of TCR signal strength is heavily dependent on the infection specific parameters such as

antigen load and inflammation. It should be noted that the rLCMV Clone-13 used in our study, which expresses WE-GP, tends to establish viremia for 20–30 days, which is somewhat less extensive than commonly observed for LCMV Clone-13 (*Fallet et al., 2016*; *Sommerstein et al., 2015*). Unspecified differences between rLCMV Clone-13 and LCMV Clone-13 might also account for some of the viral load differences we observed in the blood and kidney compared to the spleen. Although the duration of wild type Clone-13 viremia can vary considerably between animal facilities, it is possible that the Th bias we report here might have been even more pronounced if a more persistent LCMV variant had been used.

The observation that TCR signal intensity correlates with Th1 generation during acute infection is consistent with accumulating evidence that higher potency ligands increase T cell sensitivity to IL-2, which likely drives the survival and expansion of Th1 effectors (*Keck et al., 2014*; *Krishnamoorthy et al., 2017*; *Tubo et al., 2013*; *Fazilleau et al., 2009*; *Gottschalk et al., 2012*; *Ploquin et al., 2011*; *Allison et al., 2016*). This is similar to the paradigm described for Th1/Th2 cell differentiation, where stronger signals induce Th1 cells and weaker signals induce Th2 cells (*Tao et al., 1997*). Nevertheless, at very high antigen doses, T cells revert to Th2 differentiation, potentially due to the susceptibility of Th1 cells to activation induced cell death (AICD) (*Hosken et al., 1995*; *Bretscher et al., 1992*). AICD of Th1 cells might also contribute to biased Tfh generation at the higher end of TCR signal strength during Clone-13 infection (*Lohman and Welsh, 1998*; *Schorer et al., 2020*).

The shift of relatively high affinity CD4 T cells toward a Tfh cell phenotype during Clone-13 infection is well documented (*Fahey et al., 2011*; *Vella et al., 2017*; *Snell et al., 2016*). In addition to antigen persistence, however, Clone-13 presents an altered inflammatory environment which contributes to an IFN stimulated gene signature and IL-10 production by chronically activated CD4 T cells (*Crawford et al., 2014*; *Parish et al., 2014*). It is possible that the unique cytokine milieu present during Clone-13 infection cooperates with strong TCR signals to fine-tune T cell fate. For example, activation of T cells in the presence of IFNα induces T cell secretion of IL-10 which is positively regulated by TCR signal strength (*Corre et al., 2013*; *Saraiva et al., 2009*). While this may ultimately serve to limit host pathology, it may also prevent the accumulation of Th1 effectors. Consistent with this idea, blocking IFNα or IL-10 during Clone-13 infection rescues the Th1 effector compartment and improves viral control, although this likely depends on the rate of viral replication (*Teijaro et al., 2013*; *Wilson et al., 2013*; *Richter et al., 2013*). Within the same inflammatory context, weaker TCR signals might induce less T cell derived IL-10 which has been shown to impair Th1 effector cell differentiation (*Parish et al., 2014*). Of particular interest, T cell production of IL-10 during chronic infection depends on sustained, but ERK-independent TCR signals, suggesting that inflammatory versus suppressive cytokine secretion may have distinct TCR signaling requirements (*Parish et al., 2014*). Future experiments should address this by determining whether TCR signal strength contributes to cytokine production as well as cytokine susceptibility (i.e. induction of cytokine receptors) of effector cells responding during acute and chronic viral infection.

Finally, the ability of weakly activated T cells to maintain a higher proportion of Th1 effectors might ultimately contribute to viral control. The observation that the weakest Clone-13 variant, Y72F, elicited significantly more expansion than its Armstrong counterpart demonstrates that prolonged antigen presentation supports the accumulation of relatively low affinity T cells. Importantly, our study only follows the differentiation of T cells specific for a single epitope, while low affinity T cells are demonstrated to comprise up to half of the endogenous effector T cell response (*Martinez et al., 2016*). Going forward, it will be interesting to determine whether targeting the expansion of lower affinity T cells with the potential to resist functional inactivation and maintain proliferative potential will improve control of viral infection.

## Materials and methods

### Viruses

Virus rescue was performed as described previously using the pol-I/pol-II-driven reverse genetic system for LCMV (*Flatz et al., 2006*). Single amino acids changes of the GP61 epitope were introduced by site-directed mutagenesis of the previously described pI-S-WE-GP rescue plasmid (*Flatz et al., 2006*). This plasmid encodes the NP of the LCMV Armstrong strain on *cis* with the glycoprotein (GP)

of LCMV WE. In addition, the LCMV Armstrong specific D63K mutation was introduced into the GP61-coding sequence of the WE-GP gene matching the LCMV Armstrong/Clone-13 amino acid sequence of the GP61 peptides employed in the T cell activation assay. The resulting S-rescue plasmids were combined with a plasmid expressing either the Armstrong or the Clone-13 L segment in order to generate acute and chronic variants, respectively. The presence of the desired mutations in the viral genomes was verified by sanger sequencing of (reverse transcription polymerase chain reaction) RT-PCR amplicons generated with the OneStep RT-PCR-kit (Qiagen) using LCMV WE GP-specific primers (GATTGCGCTTTCCTCTAGATC and TCAGCGTCTTTTCCAGATAG). Viral RNA was extracted from cell culture supernatants using the Direct-zol RNA MicroPrep kit (Zymo Research). Virus titer was determined by immunofocus assay as described on NIH/3T3 cells (*Battegay, 1993*). To determine viral load in organs, tissues were homogenized with the TissueLyser II (Qiagen) for 2 × 1 min at 30 Hz. Recombinant LCMV Cl13 WE-GP ΔGP66 was generated with the S-plasmid from a previous publication combined with the Clone-13 L segment (*Recher et al., 2004*).

## Viral growth kinetics

To determine viral replication capacities, BHK21 cells were seeded 24 hr prior to infection with amultiplicity of infection of 0.01. Supernatant was collected at indicated time points and replaced with fresh culture medium.

## Mice and animal experiments

Mice were bred and housed under specific pathogen-free conditions at the University Hospital of Basel according to the animal protection law in Switzerland. For all experiments, male or female sex-matched mice were used that were at least 6 weeks old at the time point of infection. The following mouse strains were used: C57BL/6 CD45.2, SMARTA Ly5.1, CD74$^{-/-}$, DBA/2. Mice were injected with intraperitoneal injection of 2 × 10$^5$ FFU for Armstrong variants or via intravenous injection of 2 × 10$^6$ FFU for Clone-13 variants.

## NICD-protector

Mice were intravenously injected with 12.5 µg homemade ARTC2.2-blocking nanobody s+16 (NICD-protector) at least 15 min prior to organ harvest.

## Adoptive cell transfer

Single-cell suspensions of cells were prepared from lymph nodes by mashing and filtering through a 100 µm strainer. Naïve Smarta cells were enriched using Naïve CD4 T cell isolation kit (StemCell). 1 × 10$^4$ SMARTA Ly5.1 (2 × 10$^5$ SMARTA cells for day 4 experiments) cells were adoptively transferred into Ly5.2 recipients via intravenous injection as previously described (*Moon et al., 2009*).

## Flow cytometry

Spleens were removed and single-cell suspensions were generated by mashing and filtering the spleens through a 100 µm strainer followed by erythrocytes lysing using ammonium-chloride-potassium lysis buffer. SMARTA and endogenous LCMV-specific CD4 T cells were analyzed using IAb: NP309-328 (PE) or IAb:GP66-77 (APC) (provided by NIH tetramer core) tetramer. Following staining for 1 hr at room temperature in the presence of 50 nM Dasatinib, tetramer-binding cells were enriched using magnetic beads and counted as previously published (*Moon et al., 2009*). Surface combined with viability staining was performed for 30 min on ice. For transcription factor staining, fixation and permeabilization was performed according to the Foxp3/Transcription Factor staining kit (eBioscience). Samples were analyzed on Fortessa LSR II or Canto II cytometers (BD Biosciences) followed by data analysis with FlowJo X software (TreeStar). CD4$^+$ T cells were pregated on lymphocytes in FSC/SSC, dump$^-$, live CD4$^+$ cells and then further gated on CD44$^+$ Tetramer$^-$ to assess the CD44$^+$ compartment, CD44$^+$ CD45.1$^+$ GP66$^+$ for SMARTA, and CD44$^+$ NP309$^+$ for NP-specific cells.

## CD69 SMARTA activation assays

Serial dilutions of the GP61 wt peptide or APLs were plated. 5 × 10$^5$ Ly5.2 splenocytes and 0.5−1 × 10$^5$ Ly5.1 SMARTA cells per well were added to the dilution series, stimulated for 6, 12 (overnight), or 24 hr at 37°C, and subsequently stained and analyzed at the flow cytometer. MAR1 IFN type-I

blockade antibody (BioXcell, #BE0241) or isotype control (MOPC-21, BioXcell, #BE0083) was supplemented in the culture medium at a concentration of 20 µg/ml.

## MHC-II out-competition assays

CD74$^{-/-}$ splenocytes were cultured with a custom made GP61-FITC at a fixed concentration of $1 \times 10^{-6}$ M and various serial dilutions of GP61 wt or APLs for 4 hr at 37°C. After stimulation, the cells were stained and analyzed at the flow cytometer. The FITC-labeled GP61 peptide was custom made by Eurogentec.

## Statistical analysis

Geometric mean was used to determine the mean fluorescence intensity (MFI), and values were normalized to the mean of the control group from each experiment before data was pooled. Pooled and normalized MFIs are referred to as relative MFI. $EC_{50}$ values were calculated using a sigmoidal dose–response fit in GraphPad Prism (versions 8 and 9). For statistical analysis of one parameter between two groups, unpaired two-tailed Student's t-tests were used to determine statistical significance. To compare one parameter between more than two groups, one-way ANOVA was used followed by Turkey's post-test for multiple comparisons. p-Values are indicated on the graphs. Data was analyzed using GraphPad Prism software (version 8).

# Acknowledgements

We thank all members of the Pinschewer lab for helpful discussion, Weldy V Bonilla for technical support on determining the viral load of organ samples, and David Schreiner for editing of the manuscript. Research was supported by the Swiss National Science Foundation (SNF, grants number PP00P3_157520 to CGK and number 310030_173132 to DDP), Gottfried and Julia Bangerter-Rhyner Stiftung, Olga Mayenfisch Stiftung, the Nikolaus and Bertha Burckhardt-Bürgin Stiftung (NBB), and the Freiwillige Akademische Gesellschaft (FAG) Basel.

# Additional information

### Competing interests

Daniel D Pinschewer: a founder, consultant and shareholder of Hookipa Pharma Inc. commercializing arenavirus-based vector technology, and he is listed as inventor on corresponding patents (Replication-defective arenavirus vectors, WO 2009/083210; Anti-mycobacterial vaccines. PCT-Patent Application No. PCT/EP14/055144; Tri-segmented arenaviruses as vaccine vectors. PCT/EP2015/076458; Tri-segmented Pichinde viruses as vaccine vectors, WO 2017/198726). The other authors declare that no competing interests exist.

### Funding

| Funder | Grant reference number | Author |
| --- | --- | --- |
| Swiss National Science Foundation | PP00P3_157520 | Carolyn G King |
| Swiss National Science Foundation | 310030_173132 | Daniel D Pinschewer |
| Gottfried and Julia Bangerter-Rhyner Foundation | | Carolyn G King |
| Nikolaus and Bertha Burckhardt-Bürgin Stiftung | | Marco Künzli |
| Freiwillige Akademische Gesellschaft | | Marco Künzli |
| Olga Mayenfisch Stiftung | | Carolyn G King |

The funders had no role in study design, data collection and interpretation, or the decision to submit the work for publication.

## Author contributions
Marco Künzli, Formal analysis, Funding acquisition, Investigation, Visualization, Methodology, Writing - original draft, Writing - review and editing; Peter Reuther, Investigation, Methodology, Writing - review and editing; Daniel D Pinschewer, Funding acquisition, Writing - review and editing, provided advice on experimental design; Carolyn G King, Conceptualization, Formal analysis, Funding acquisition, Writing - original draft, Writing - review and editing

## Author ORCIDs

Marco Künzli (iD) https://orcid.org/0000-0001-7699-779X
Carolyn G King (iD) https://orcid.org/0000-0001-6059-323X

## Ethics
Animal experimentation: Mice were handled according to the animal protection law in Switzerland. The protocol was approved by the veterinary authority (Basel-Stadt) under the animal license number 2991.

## Decision letter and Author response
Decision letter https://doi.org/10.7554/eLife.61869.sa1
Author response https://doi.org/10.7554/eLife.61869.sa2

# Additional files
## Supplementary files
• Supplementary file 1. Altered peptide ligands (APLs) with altered potential to activate SMARTA and corresponding $EC_{50}$ values.

• Transparent reporting form

## Data availability
Source data files have been provided for Figures 1-4 and Figure supplements.

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
