## [Decision Letter]

**Acceptance summary:**

The manuscript demonstrates that TCR signal strength can dominantly instruct the development of Th1 and T follicular helper cells across distinct infectious contexts. It involves a considerable amount of work resulting in the generation of LCMV variant strains to examine the impact of TCR signal strength on CD4 T cells responding during acute and chronic viral infection.

**Decision letter after peer review:**

Thank you for submitting your article "Opposing effects of T cell receptor signal strength on CD4 T cells responding to acute versus chronic viral infection" for consideration by *eLife*. Your article has been reviewed by two peer reviewers, and the evaluation has been overseen by a Reviewing Editor and Tadatsugu Taniguchi as the Senior Editor. The reviewers have opted to remain anonymous.

The reviewers have discussed the reviews with one another and the Reviewing Editor has drafted this decision to help you prepare a revised submission.

Both reviewers appreciated your attempts to determine whether TCR signal strength can dominantly instruct the development of Th1 and T follicular helper cells across distinct infectious contexts and the large efforts you invested (1) in the development of a wide panel of altered peptide ligands for the LCMV-derived GP61 peptide and (2) the generation of LCMV variant strains to examine the impact of TCR signal strength on CD4 T cells responding during acute and chronic viral infection. However, they both came to the same conclusion that altering the virus could have profound impacts on the infection kinetics, tropism and antigen load. It might thus lead to over interpretation of the data. Along that line, it would be important to show virus titers in the spleen and blood over time to demonstrate that the virus variants have similar infection and persistence kinetics as the parenteral line. It is also suggested to mitigate the conclusions related to the "Goldilocks model".

Reviewer #1:

In the manuscript by Kunzli et al., the authors attempt to determine how TCR single strength alters CD4 T cell fate in acute versus chronic infection. While this is an interesting question, there are some major concerns about the interpretation of the data. First, the amino acid mutations introduced into the GP protein altered the fitness of 3/5 of the variants such that only 2 were viable. There is not enough evidence presented in Figure 1D (day 3 viral load- no stats) to demonstrate that there are not in vivo alterations of fitness or viability in the mutant strains they used. Far more detailed measurements of viral load, tissue tropism and clearance kinetics need to be assessed. Second, the interpretation that an overnight stimulation with total splenocytes leading to CD69 up-regulation is a direct measurement of TCR signal strength is a gross oversimplification. CD69 is up-regulated in response to TCR as well as Type 1 interferon signaling. As discussed at length in Andargachew et al., 2018, functional readouts of T cell activity are not necessarily a direct correlate of TCR affinity (as measured by 2D affinity). In addition the assessment of T cell fate is primarily done at 10 days post-infection but what are the overall kinetics of the T cell responses? Do the GCs form with similar kinetics? Taking a snapshot of one timepoint may not be representative of differences in fate determination that may occur at different timepoints. These are interesting preliminary results, but far more work supporting the use of this system needs to be done.

Reviewer #2:

Künzli et al. address how differences in the strength of stimulating CD4 T cells impacts their differentiation potential and their ability to generate Th1 versus Tfh cells. To this end they undertook the enormous effort of generating altered peptide ligand expressing LCMV. This afford should be highly appreciated.

The results for the CD4 T cells resemble a trend that was already seen for CD8 T cells exposed to APL expressing LCMV, which was that signal strength correlated with phenotypic features of exhausted T cells. However, what is largely unexpected is the differential response of the Tfh versus Th1 differentiation in acute versus chronic infection.

Overall the manuscript is very well written and provides insightful observations. I have only a few relevant points.

– I am not convinced by the conclusion that “too much or not enough” induces Tfh cells. Just because chronic and acute infection favor opposite outcome does in my opinion sufficiently support this conclusion. Maybe T cell respond differentially in both types of infection. What would be the functional benefits from such a system? I would suggest to amend this conclusion.

– It would be important to show virus titers in the spleen and blood over time to sure that the variants have similar infection kinetics as the parenteral line.

– What remains still puzzling is the difference pattern of Tfh formation in the acute or chronic infection. It would be interesting to see more mechanistic support for this observations

---

## [Author Response]

Both reviewers appreciated your attempts to determine whether TCR signal strength can dominantly instruct the development of Th1 and T follicular helper cells across distinct infectious contexts and the large efforts you invested (1) in the development of a wide panel of altered peptide ligands for the LCMV-derived GP61 peptide and (2) the generation of LCMV variant strains to examine the impact of TCR signal strength on CD4 T cells responding during acute and chronic viral infection. However, they both came to the same conclusion that altering the virus could have profound impacts on the infection kinetics, tropism and antigen load. It might thus lead to over interpretation of the data. Along that line, it would be important to show virus titers in the spleen and blood over time to demonstrate that the virus variants have similar infection and persistence kinetics as the parenteral line. It is also suggested to mitigate the conclusions related to the "Goldilocks model".

To address the main concern of both reviewers, we determined the infection kinetics and tropism of recombinant viruses by measuring viral load in the spleen, blood, kidney, and liver at days 4, 7, 14 and 21 after infection (please note: as LCMV Armstrong is cleared by day 7 (*1*), these experiments were only carried out for Clone-13 viruses). Here we observed that GP61 wt as well as V71S and Y72F variants persisted in the blood, spleen and kidney of C57BL/6 mice throughout the 21-day observation period, with comparable viral titers in blood and organs at most time points. These findings render it unlikely that differential intrinsic fitness or organ distribution of the variant viruses account for the observed impact on SMARTA CD4 T cell differentiation.

It is unclear why the viral loads in the kidneys at day 7 are slightly different than the data presented in Figure 3—figure supplement 4. One possibility is that the host mice for several revision experiments were ordered from an outside vendor due to SARS-CoV2 breeding constraints. Nevertheless, viral load in the spleen was consistent across wild type and variant strains and the impact of TCR signal strength on SMARTA T cell differentiation was maintained.

We additionally measured viral load kinetics in the spleen, kidney, and liver (day 21) as well as the blood (time course) in DBA/2 mice which are unable to present the GP66 epitope. This allowed us to compare viral fitness in the absence of endogenous CD4 T cell responses which might contribute to differences in viral clearance. Here again we observed comparable viral titers across all organs and variant viruses.

We have revised the manuscript to incorporate these new data (Figure 1E and Figure 1—figure supplement 3).

Reviewer #1:In the manuscript by Kunzli et al., the authors attempt to determine how TCR single strength alters CD4 T cell fate in acute versus chronic infection. While this is an interesting question, there are some major concerns about the interpretation of the data. First, the amino acid mutations introduced into the GP protein altered the fitness of 3/5 of the variants such that only 2 were viable. There is not enough evidence presented in Figure 1D (day 3 viral load- no stats) to demonstrate that there are not in vivo alterations of fitness or viability in the mutant strains they used. Far more detailed measurements of viral load, tissue tropism and clearance kinetics need to be assessed.

Please see response to main comment.

Second, the interpretation that an overnight stimulation with total splenocytes leading to CD69 up-regulation is a direct measurement of TCR signal strength is a gross oversimplification. CD69 is up-regulated in response to TCR as well as Type 1 interferon signaling. As discussed at length in Andargachew et al., 2018, functional readouts of T cell activity are not necessarily a direct correlate of TCR affinity (as measured by 2D affinity).

We agree that this is an important point raised by the reviewer, particularly as high and low potency ligands can have a similar TCR affinity (*2*). For this reason, and in the absence of direct affinity measurements such as 2D affinity or surface plasmon resonance, we used the words “TCR signal strength” in place of “affinity” throughout the manuscript. To further address the reviewer’s comment, we measured CD69, IRF4 and CD25 expression on SMARTA T cells activated with variant peptides in the presence of an anti-IFNAR blocking antibody or an isotype control. After 6 hours of stimulation, CD69 expression was similarly induced in control and anti-IFNAR treated cultures with no change to the TCR signal strength hierarchy. Expression of both IRF4 and CD25 (at 24 hours) was also induced according to TCR signal strength, although here we observed slightly decreased expression of these markers in the presence of IFNAR blockade at lower doses of variant peptides. We have added these data to the revised manuscript (Figure 1—figure supplement 2).

In addition the assessment of T cell fate is primarily done at 10 days post-infection but what are the overall kinetics of the T cell responses? Do the GCs form with similar kinetics? Taking a snapshot of one timepoint may not be representative of differences in fate determination that may occur at different timepoints.

To assess the kinetics of T cell responses we measured the number of SMARTA T cells at days 4 and 10 after Armstrong infection and days 4, 7 and 14 after Clone-13 infection. Data for day 4 are analyzed separately as we transferred 20x more SMARTA T cells in order to be able to perform phenotypic analysis at this early time point. At all timepoints after either acute or chronic viral infection, we observed that the expansion hierarchy correlates with TCR signal strength (Figure 2—figure supplement 1 and Figure 3—figure supplement 6).

To assess the early generation of Th1 effectors, we measured CD25 expression at day 4 after infection (*3*). As expected, CD25 was more highly expressed by SMARTA T cells responding to GP61 wt Armstrong compared to V71S (Figure 2—figure supplement 2). In contrast, CD25 was only weakly expressed on SMARTA T cells responding to wild type or Clone-13 variants making it an unreliable marker for early Th1 cell differentiation (Figure 3—figure supplement 7). In place of CD25 we used Ly6c expression as a readout for early Th1 effectors (*4*). Similar to our observations at day 7 after infection, Ly6c negatively correlated with TCR signal strength (Figure 3—figure supplement 7). Higher expression of Ly6C by weakly activated T cells was maintained at day 14 p.i. although the difference between V71S and Y72F variants was no longer apparent at this time point (Figure 3—figure supplement 8).

To determine the kinetics of GC formation we measured the proportion of GL7^+^Fas^+^ B cells at days 4 (Armstrong variants) and at day 4 and 14 (Clone-13 variants) after infection. At day 4, GC B cells are essentially absent in both Armstrong and Clone-13 variants, suggesting that variant viruses do not form GCs earlier than wild type viruses (Figure 2—figure supplement 3 and Figure 3—figure supplement 9). We also didn’t observe any differences at day 7 or 14 p.i., indicating that GC kinetics and magnitudes are comparable among the Clone-13 variants (Author response 1, Figure 3—figure supplement 9). It is important to note, however, that while the experimental system we are using is optimal for studying T cell intrinsic differentiation it does not preclude the contribution of non-SMARTA T cells to GC formation. To address the impact of SMARTA T cells on GC B cell formation would require either TCR-restricted or CD4 T cell deficient hosts which is beyond the scope of this manuscript.

**Author response image 1. sa2fig1:** Germinal center B cell differentiation 7 days post LCMV Clone-13 variant infection. Proportion and numbers of GL7+FAS+ B cells 7 days p.i. Data are pooled from n = 2 independent experiments with 8 samples per group. Bars represent the mean and symbols represent individual mice. Significance was determined by unpaired two-tailed Student’s t-tests.

These data and adjusted text have been added to the revised manuscript.

These are interesting preliminary results, but far more work supporting the use of this system needs to be done.Reviewer #2:Künzli et al. address how differences in the strength of stimulating CD4 T cells impacts their differentiation potential and their ability to generate Th1 versus Tfh cells. To this end they undertook the enormous effort of generating altered peptide ligand expressing LCMV. This afford should be highly appreciated.The results for the CD4 T cells resemble a trend that was already seen for CD8 T cells exposed to APL expressing LCMV, which was that signal strength correlated with phenotypic features of exhausted T cells. However, what is largely unexpected is the differential response of the Tfh versus Th1 differentiation in acute versus chronic infection.Overall the manuscript is very well written and provides insightful observations. I have only a few relevant points.– I am not convinced by the conclusion that “too much or not enough” induces Tfh cells. Just because chronic and acute infection favor opposite outcome does in my opinion sufficiently support this conclusion. Maybe T cell respond differentially in both types of infection. What would be the functional benefits from such a system? I would suggest to amend this conclusion.

Please see response to main comment.

– It would be important to show virus titers in the spleen and blood over time to sure that the variants have similar infection kinetics as the parenteral line.

Please see response to main comment.

– What remains still puzzling is the difference pattern of Tfh formation in the acute or chronic infection. It would be interesting to see more mechanistic support for this observations

We tried to address this question by dissecting the impact of antigen dose versus TCR signal strength during Clone-13 infection. We examined SMARTA T cell differentiation in a mixed infection setting 7 days post infection:

high antigen dose, strong TCR signal 100% Clone-13 GP61 wt

low antigen dose, strong TCR signal 10% Clone-13 GP61 wt + 90% Clone-13 ΔGP66

high antigen dose, weak TCR signal 100% Clone-13 Y72F

low antigen dose, weak TCR signal 10% Clone-13 Y72F + 90% Clone-13 ΔGP66

We found that lowering the antigen dose decreased the expansion of SMARTA cells receiving strong TCR signal (GP61 wt) but increased the proportion of Th1 effectors 7 days post infection. In contrast, the differentiation outcome of SMARTA cells receiving weak TCR signal was independent of the antigen load. However, we acknowledge that this is only a snapshot of the CD4 response to less antigen and that over time one of the viruses from the mixed setting could outcompete the other at later time points. We added these data to the manuscript and modified the manuscript.

References

1. E. J. Wherry, J. N. Blattman, K. Murali-Krishna, R. van der Most, R. Ahmed, Viral persistence alters CD8 T-cell immunodominance and tissue distribution and results in distinct stages of functional impairment. J Virol 77, 4911-4927 (2003).

2. G. J. Kersh, E. N. Kersh, D. H. Fremont, P. M. Allen, High- and low-potency ligands with similar affinities for the TCR: the importance of kinetics in TCR signaling. Immunity 9, 817-826 (1998).

3. J. P. Snook, C. Kim, M. A. Williams, TCR signal strength controls the differentiation of CD4(+) effector and memory T cells. Sci Immunol 3, (2018).

4. H. D. Marshall et al., Differential expression of Ly6C and T-bet distinguish effector and memory Th1 CD4(+) cell properties during viral infection. Immunity 35, 633-646 (2011).